# The pattern of lung function tests in children with sickle cell disease: A case-control study

**Abinaya Kannan, Gaurav Sarnaik, Nikita Agarwal, Atul Jindal** *

Department of Pediatrics, All India Institute of Medical Sciences, Tatiband, Raipur, Chhattisgarh, India

* dratuljindal@gmail.com

## Background

Sickle cell disease (SCD) is major inherited disease linked to a pro-inflammatory state, with a widespread involvement seen in all organ systems with lungs being no different. This research aims to analyze pulmonary function tests and fractional exhaled nitric oxide (FeNO) levels of children diagnosed with SCD and comparing them with healthy controls.

## Methods

The study involved 100 children with SCD in stable state of health, without pain, crises, or acute illnesses for at least 1-month, and aged between 6–20 years and 100 age- and height-matched controls. Those with spinal deformities, acute or chronic cardiorespiratory symptoms, and cigarette smoking (in parents) were excluded. Measurements of forced vital capacity(FVC) and forced expiratory volume in one second(FEV1) were conducted using an automated single-breath vitalograph. FeNO was recorded using a hand-held device(NIOX MINO). Data werecollected and analyzed

## Results

Children with SCD had significantly lower pulmonary function values compared to controls: FEV1 median difference: 33.5% (95% CI: 27.2–38.0; p < 0.0001), FVC: 25.4% (95% CI: 30.0–32.25; p < 0.0001), FEV1/FVC: 0.088 (95% CI: 0.075–0.083; p < 0.0001), Peak Expiratory Flow Rate (PEFR): 24.8% (95% CI: 16.38–33.8; p < 0.0001), FeNO: 8.17 ppb (95% CI: 5.77–12.65; p < 0.0001). Pulmonary function test (PFT) abnormalities were associated with younger age (p = 0.0022). Age (p = 0.0011) was significantly associated with PFT severity, while blood transfusion frequency, and fractional exhaled nitric oxide (FeNO) levels were not. Increased weight (p = 0.001) and longer duration of hydroxyurea (p = 0.011) were associated with improved PFT severity (based on FEV1 z-scores).

## Conclusions

Children with SCD often exhibit restrictive, obstructive, or mixed pulmonary function patterns. FeNO levels donot correlate with PFT severity.

**Data availability statement:** All relevant data are within the manuscript and its Supporting Information files.

**Funding:** The author(s) received no specific funding for this work.

**Competing interests:** The authors have declared that no competing interests exist.

## Introduction

In the urban areas of Chhattisgarh, India, sickle cell disease (SCD) occurs at a prevalence of 6.6 cases for every 1,000 individuals. This makes it a major inherited disease in the region, similar to trends seen worldwide [1].Numerous acute conditions are linked to SCD, including vaso-occlusive crises (VOC), acute chest syndrome (ACS), aplastic crises, and hemolytic crises, with VOC and ACS being the most common. [2,3].Patients who experience repeated episodes of ACS may develop patchy lung fibrosis, mainly found in the lower lobes of the lungs [4].Sickle cell lung disease is a complex condition involving lower airway disease, sleep-disordered breathing, and environmental factors. Understanding its presentation, pathophysiology, and diagnostic approaches is crucial for effective management. Collaborative efforts among healthcare providers, researchers, and policymakers are essential [5].

The pathophysiological processes underlying atopic asthma are characterized by the upregulation of T-helper 2 (Th2) cytokines, inflammation driven by mast cells and eosinophils, and heightened activity of inducible nitric oxide synthase (iNOS) and arginase within the airway epithelium. These factors contribute to obstructive changes and airway hyper-responsiveness (AHR). However, the precise mechanisms leading to AHR, as well as obstructive and restrictive lung diseases in SCD, remain poorly understood. It is established that SCD is linked to a pro-inflammatory state, with an intensified inflammatory response observed during VOC. Potential mechanisms for pulmonary dysfunction in SCD patients include hemolysis-induced acute-on-chronic inflammation and dysregulated metabolism of nitric oxide (NO) [6]. Consequently, markers of arginine metabolism may serve to distinguish between different asthma phenotypes in pediatric patients with SCD [7].

In comparison to patients with atopic asthma, individuals with SCD exhibit elevated levels of inflammatory markers associated with Th-1, Th-2, and monocytic pathways [8]. FeNO levels serve as a biomarker for eosinophilic airway inflammation in children with asthma. However, studies in children with SCD have produced conflicting reports [9–11].Individuals with SCD may face an increased risk of developing chronic lung conditions due to the dysregulation of NO metabolism stemming from hemolysis [12].

This study examines the PFT and FeNO results of children with SCD (ages 6–20 years) in comparison to age-, sex-, and height-matched individuals without SCD.

## Materials and methods

This case-control study was carried out between April 2017 and October 2017 at the outpatient department of the Department of Pediatrics of our institution, following the approval of the Institutional Ethics Committee (AIIMSRPR/IEC/2017/074). The research adhered to the principles outlined in the Declaration of Helsinki. Informed written consent was obtained from the parents or guardians, as appropriate, prior to the enrollment of their children or wards in the study. Participants included patients with SCD who were in a stable state of health, without pain, crises, or acute illnesses for at least one month, and aged between 6 and 20 years, classified as cases. This age group was chosen to facilitate early pick-up of the disease. Healthy, non-asthmatic, non-SCD children, matched for age, height, and sex, were recruited as

controls from government schools and colleges through a random multi-stage sampling method. Exclusion criteria encompassed children with spinal deformities, any acute or chronic cardio-respiratory conditions that could affect lung function, and a history of cigarette smoking among the subjects, controls, or their household members. The case group comprised 100 patients, who were compared with 100 control participants, with both groups matched for age, height, and gender. The sample size was determined based on recruitment ability within the timeframe of the study.

Measurements of forced vital capacity (FVC) and forced expiratory volume in one second (FEV1) were conducted utilizing an automated single-breath vitalograph. The assessment of peak expiratory flow rate (PEFR), FEV1, and FVC adhered to the standard protocols established by the American Thoracic Society (ATS) guidelines [13]. The tests met acceptability and reproducibility prior to being included in this analysis. At least three acceptable and reproducible maneuvers were performed and the best values were considered. The resultant pulmonary function tests (PFTs) were categorized according to the ATS classification system [14].

An obstructive pattern is characterized by a reduced FEV1/FVC ratio, below LLN (lower limit of normal). A restrictive pattern is indicated by an FVC that falls below the LLN of predicted FVC for all patients. If both the FEV1/FVC ratio and FVC are diminished below the LLN, the patient was characterized as having a mixed pattern. For grading severity z-score of FEV1 was used as Mild: −1.65 to −2.5, Moderate: −2.51 to −4.0 and Severe: <−4.1 [15].

FeNO was recorded using a quality assured and calibration-free hand-held device (NIOX MINO) for the measurement of FENO according to ATS's guidelines of 2011 [16].

The technicians conducting the PFTs and FeNO estimation were blinded to the SCD status of the child. The analysis could not be blinded as details of disease were collected for severity-related assessments.

Information on the history of blood transfusions and the duration of hydroxyurea administration was gathered from medication records for the cases.

All data were coded, entered, and subsequently analyzed utilizing SPSS version 25. For the variables including PEFR, FEV1, and FVC Mann-Whitney test was applied as data was skewed. The chi-square test facilitated the comparison of variables such as age and gender between the SCD patients and the control group. Additionally, Spearman's correlation was calculated among the variables. Age was used as a substitute of duration of illness (as SCD is a genetic disorder) and it was correlated using Kruskal-Wallis Test to the type of PFT abnormality. A Spearman Rank Co-relation was done to compare age (hence duration of illness) to severity of PFT dysfunction. A p-value of less than 0.05 was deemed statistically significant.

## Results and discussion

A total 150 children with SCD were screened. Of the 150, 25 had a crisis at the time of or within one month of presentation and were therefore excluded. Ten children were being treated with inhaled bronchodilators therefore were excluded. Seven children had spinal deformities and eight children's parents refused to consent to the study. Therefore, thestudy comprises 100 individuals diagnosed with SCD, which is compared to a control group of 100 healthy individuals, matched for age, height, and sex. The flow of the study is depicted in Fig 1. The demographic characteristics are presented in Table 1.

Given that the ages have been successfully matched, there is no age difference between the two cohorts. The median age for both groups was 13.5 years (Inter-quartile range (IQR) 11–17 years). The median height was recorded at 144.5 cm (IQR 130–157.13 cm) for the group with patients diagnosed with SCD, while the control group had a median height of 146 cm (IQR 130.5–156 cm). Additionally, females constituted 39% of the population in each group.

The details of FEV1, FVC, FEV1/FVC ratio, PEFR and FeNO are tabulated in Table 2 and visually represented in Fig 2.

The median difference of FEV1 is 33.5 (95% CI: 27.2–38) (p<0.0001), FVC is 25.4 (95% CI: 30–32.25) (p<0.0001), FEV1/FVC is 0.088 (95% CI:0.075–0.083) (p<0.0001), PEFR is 24.8 (95% CI: 16.38–33.8) (p<0.0001). Median difference of FeNO is 8.17ppb (95% CI: 5.77–12.65) (p<0.0001)

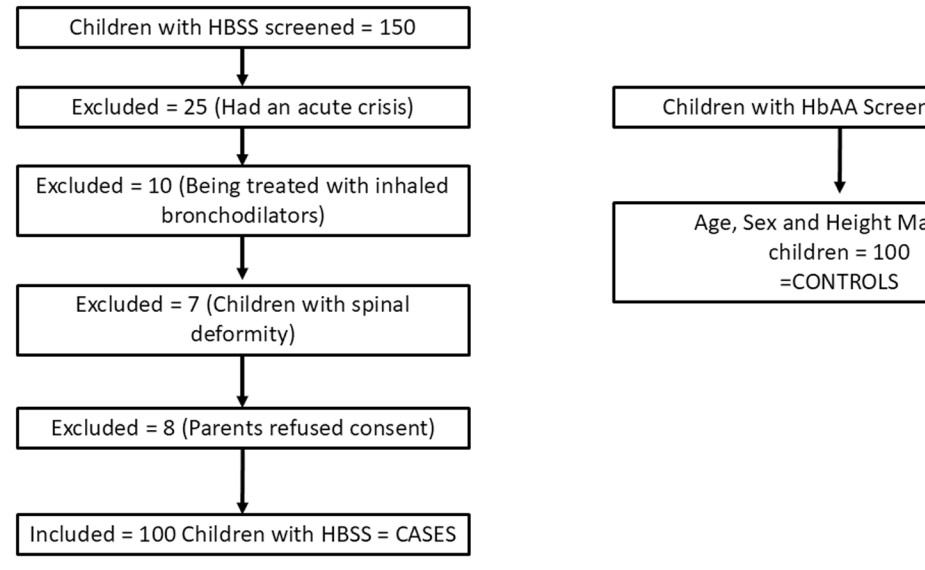

**Fig 1. Study flow.**

**Table 1. Demographic details of cases and controls.**

|  | Cases | | Controls | | p-value |
|---|---|---|---|---|---|
|  | Median (IQR) | Range | Median (IQR) | Range |  |
| Age (years) | 13.5 (11 - 17) | 6 - 20 | 13.5 (11 - 17) | 6 - 20 | 1.00 |
| Height (cm) | 144.5 (130 - 157.1) | 107.4 - 181 | 146 (130.5-156) | 110-181 | 0.936 |
| Weight (Kg) | 37 (24 - 43) | 15 - 61 | 33.05(22.95-42.75) | 14.8-68 | 0.963 |

**Table 2. Pulmonary function and FeNOTest values of cases and controls.**

|  | Cases | | Controls | | p-value |
|---|---|---|---|---|---|
|  | Median (IQR) | Range | Median (IQR) | Range |  |
| FEV1% | 66.5 (60–75.5) | 39–96 | 100 (98–102.7) | 93.5–109.6 | <0.0001 |
| FVC % | 74.5 (65.5–74.5) | 41–105 | 99.9 (97.75–104.5) | 86.7–110.7 | <0.0001 |
| FEV1/FVC | 0.820 (0.800–0.855) | 0.740–1.00 | 0.908 (0.883–0.930) | 0.743–0.987 | <0.0001 |
| PEFR% | 75 (65 - 85) | 35 - 130 | 99.80 (98.80 - 101.38) | 95.4 - 105.2 | <0.0001 |
| FeNO (ppb) | 10 (5 - 13) | 5 - 39 | 18.17 (17.65 - 18.77) | 15.77 −20.57 | <0.0001 |

All controls had normal PFT.The details of abnormality in PFTs of cases are summarized in Table 3.In the current study, only 11% of children and adolescents diagnosed with SCD showed normal PFTs. Among those with abnormal results, 32% exhibited a mixed pattern, while 28% presented a restrictive pattern and 29% demonstrated an obstructive pattern.

The odd ratio of having an abnormal PFT in children with SCD was 1564.3043 (95% CI: 90.8698 to 26929.1677).The corrected α using Bonferroni correction method is 0.008333, arrived at by making 6 comparisons. It was found that the median age was statistically more for those with mixed (Median (IQR) – 17 (11–17.5)) and restrictive PFTs (Median (IQR) – 17 (12–19)). The number of blood transfusion, duration of hydroxyurea use (in months) and level of FeNO were not significantly different for each type of abnormality as stated in Table 4.The heat chart of FeNO Spearman Correlation is depicted in Fig 3.

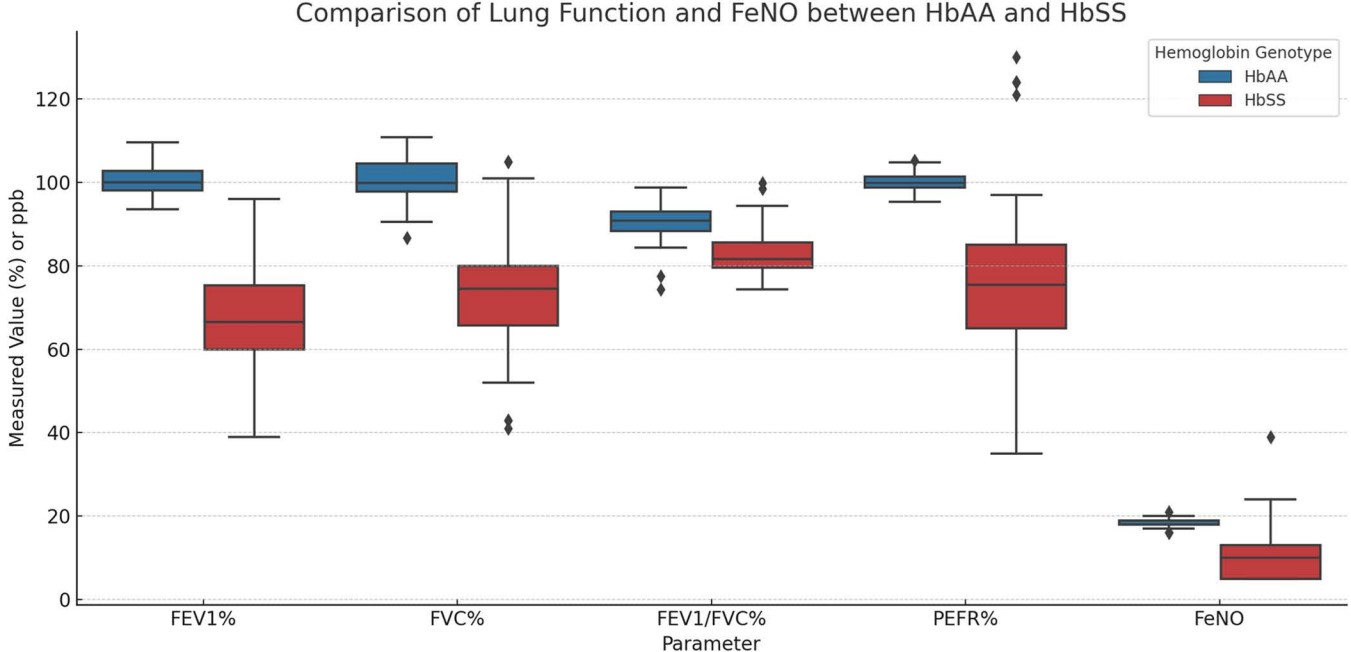

**Fig 2. Distribution of FVC, FEV1, FEV1/FVC ratio, PEFR and FeNO.**

**Table 3. Pattern and severity of PFT amongst cases.**

| Type | N | Severity | n/N (%) |
|---|---|---|---|
| **Mixed** | 32 | Mild | 8/32 (25%) |
| | | Moderate | 15/32(46.9%) |
| | | Severe | 9/32 (28.1%) |
| **Restrictive** | 28 | Mild | 3/28(10.7%) |
| | | Moderate | 19/28(37.9%) |
| | | Severe | 6/28(21.4%) |
| **Obstructive** | 29 | Mild | 29/29 (100%) |
| **Normal** | 11 | | |

As depicted, FeNO was found to be conversely related to the number of blood transfusions the child had received. On multivariate regression analysis, age and weight were significantly associated with severity of illness (FEV1 z-scores taken as a proxy for severity) as depicted in Table 5.

Older age is associated with worsening of the dependent variable (FEV1 z-scores), and this is statistically significant ($p = 0.0011$). Weight is associated with improvement in the dependent variable (FEV1 z-scores) and this is statistically significant (p = 0.001). FEV1 z-scores were better for more duration of hydroxyurea (in months) administration and this was also statistically significant (p = 0.0107).

## Discussion

In the current study, only 11% of children and adolescents diagnosed with SCD showed normal PFTs. Among those with abnormal results, 32% exhibited a mixed pattern, while 28% presented a restrictive pattern and 29% demonstrated an

**Table 4. Kruskal wallis test results for Type of PFT dysfunction.**

| | Mixed (Median, IQR) | Obstructive (Median, IQR) | Restrictive (Median, IQR) | Normal (Median, IQR) | p-value | Corrected α (Bonferroni) | Effect size |
|---|---|---|---|---|---|---|---|
| **Age (in years)** | 17 (11–17.5) | 12 (9.5–15) | 17 (12–19) | 14 (8–18) | 0.02079 | 0.008333 | 0.07 |
| **Duration of Hydroxyurea (in months)** | 0 (0–4.5) | 0 (0–42) | 0 (0–12) | 4 (0–60) | 0.4841 | | −0.0058 |
| **Number of blood transfusions** | 3 (0–12) | 4.5 (0.5–48) | 4 (0–6) | 2 (0–26) | 0.5575 | | −0.01 |
| **FeNO (ppb)** | 8 (6–13.5) | 10 (5–12) | 10 (5.5–12) | 12.5 (6–17) | 0.3662 | | 0.0018 |

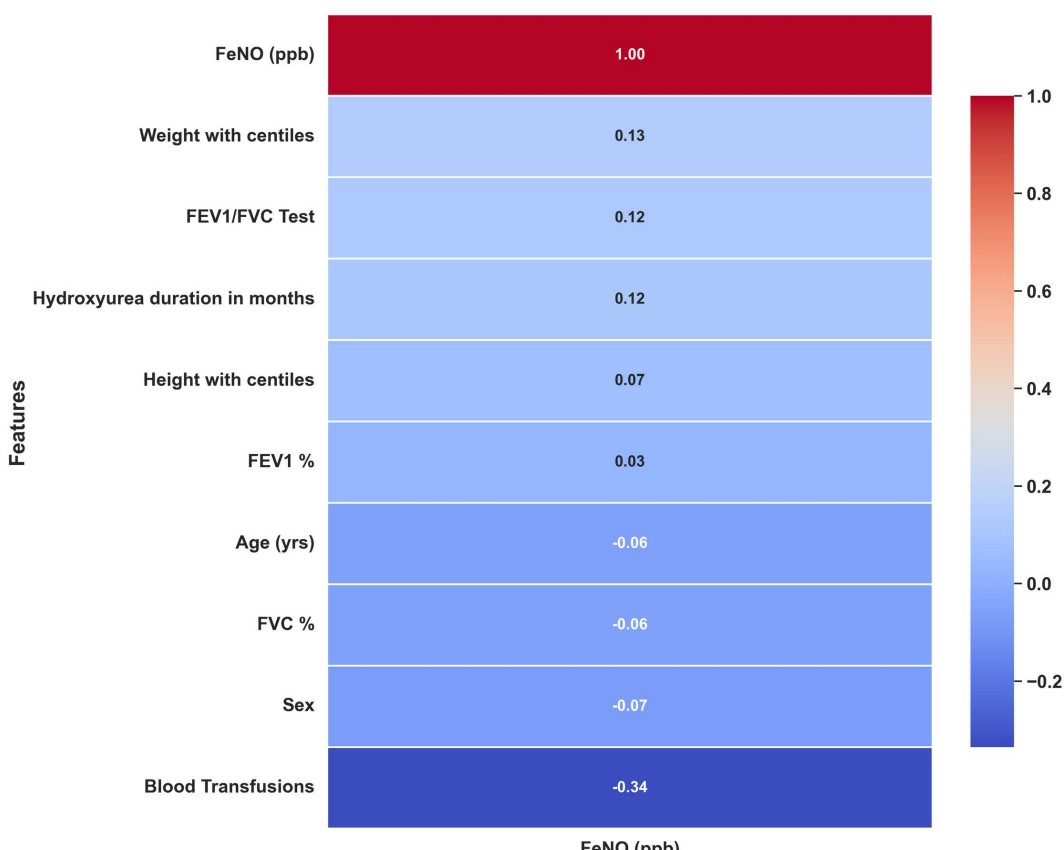

**Fig 3. Heat-chart of FeNO spearman correlation.**

**Table 5. Multivariate regression analysis.**

| | Coeff | 95% CI | t-stat | p-value |
|---|---|---|---|---|
| **Constant** | −2.962 | −6.136092 to 0.2114 | −1.857 | 0.0669 |
| **Age (in years)** | −0.085 | −0.134629 to −0.035 | −3.371 | 0.0011* |
| **Height (in cms)** | −0.01 | −0.0405282 to 0.0204 | −0.655 | 0.5143 |
| **Weight (in Kgs)** | 0.0794 | 0.0329744 to 0.1258 | 3.4025 | 0.001* |
| **Hydroxyurea (duration in months)** | 0.0103 | 0.0024549 to 0.0181 | 2.613 | 0.0107* |
| **Transfusion (Nos.)** | 0.0011 | −0.00746164 to 0.0097 | 0.2572 | 0.7977 |
| **FeNO (ppb)** | 0.0165 | −0.0308617 to 0.0638 | 0.6918 | 0.491 |

obstructive pattern. These contrasts sharply with the control group, which had uniformly normal PFTs. Additionally, significant differences were noted between the cases and controls regarding FEV1, FVC, FEV1/FVC ratio, PEFR, and FeNO levels. In multivariate analysis, older age is associated with worsening of the dependent variable (FEV1 z-scores), and this is statistically significant ($p = 0.0011$). Weight is associated with improvement in dependent variable (FEV1 z-scores) and this is statistically significant ($p = 0.001$). FEV1 z-scores were higher for longer duration of hydroxyurea (in months) administration and this was also statistically significant ($p = 0.0107$).

In 2020, in a study done by Biltagi et al, revealed pulmonary function tests conducted on children and adolescents with SCD indicated that FEV1, FVC, FEV1/FVC, total lung capacity (TLC), diffusing capacity of the lungs for carbon monoxide (DLCO), and corrected DLCO (DLCOc) were significantly reduced in patients with sickle cell disease compared to healthy control subjects [17].They also found PFT results to be abnormal in 51.1% of children diagnosed with SCD, with approximately 15% exhibiting a restrictive pattern of pulmonary function. The overall rate of abnormal findings (an obstructive, restrictive or mixed pattern on PFT) was 89%, significantly exceeding that of the Biltagi cohort.

In 2021, Taksande et al. conducted a systematic review and meta-analysis to examine PFT abnormalities in children diagnosed with SCD. This meta-analysis encompassed nine studies, which collectively involved 788 children with SCD and 1,101 healthy controls. The assessment of pulmonary function was carried out through spirometry, lung volume measurements, and gas diffusion evaluations. The findings revealed that SCD is linked to significant mean reductions in FEV1 and FVC, with differences of −12.67% and −11.69%, respectively, highlighting a considerable decline in lung function among patients with SCD [18]. This meta-analysis is in sync with the findings in our study where reduction in FEV1 and FVC were significant. We however did not do definitive diagnosis for restriction with lung-volume studies, so the findings of our study are mainly 'pattern'-based.

Research conducted by Arteta et al. identified obstructive pulmonary function as the predominant abnormality in children with SCD, correlating it with advanced age, a history of asthma or wheezing, and elevated Lactate Dehydrogenase (LDH) levels [19]. In our investigation, we did not measure LDH and excluded children with a history of asthma. Nevertheless, we observed that 29% of the children exhibited an obstructive pattern in their PFTs. Like the study by Arteta [19], we also found that higher age was associated with worse PFTs.

A similar study was done by Adekile et al on Kuwaiti children [20]. The research indicated that the correlation coefficients between FEV1 and various anthropometric measurements, as well as hemolysis markers, demonstrated a significant relationship with height ($r = 0.9$, $p < 0.001$) and weight ($r = 0.8$, $p < 0.001$). Our findings were consistent, showing that weight was notably linked to improved FEV1 z-scores. In our study children who were treated with hydroxyurea for an extended duration exhibited enhanced FEV1 z-scores. Hydroxyurea is expected to increase fetal Hemoglobin (Hb) in SCD, likely reducing hemolysis [21].This does not align with the findings of the study of Adekile et al [20] as they did not demonstrate any significant correlation with Hb, fetal Hb, reticulocytes, total bilirubin, or LDH.

The research conducted by Girgis et al. assessed FeNO levels in 44 adults with SCD and 30 healthy individuals to explore the possible influence of airway NO deficiency on pulmonary vaso-occlusion associated with SCD. The findings revealed that FeNO concentrations were markedly lower in the SCD cohort compared to the healthy controls ($14.8 \pm 8.4$ ppb vs. $24.9 \pm 13.5$ ppb, $P < 0.001$). Furthermore, SCD patients experiencing dyspnea exhibited significantly diminished FeNO levels relative to those without dyspnea ($10.1 \pm 5.7$ ppb vs. $19.6 \pm 8$ ppb, $P < 0.001$). Additionally, individuals with a history of ACS had lower FeNO values compared to those who had not experienced ACS episodes ($13.0 \pm 8.3$ ppb vs. $18.4 \pm 7.6$ ppb, $P < 0.05$) [22].While our study is constrained by insufficient data regarding dyspnea and the frequency of ACS episodes, we also observed that children with SCD had significantly lower FENO levels compared to their healthy counterparts.

Radhakrishnan et al assessed the levels of NO in the alveoli and airways of children with SCD by evaluating various flows of FENO, bronchial NO flux (J'awNO), and alveolar NO concentration (CaNO) [11].They found that the children with SCD exhibited FeNO levels within the normal range at a flow rate of 50 mL/sec; however, FeNO was significantly higher

across all flow rates when compared to healthy controls (P = 0.03). The authors concluded that lower airway NO levels are elevated in children with SCD, and the increased J'awNO may suggest a dysregulation in NO metabolism or the presence of subclinical airway inflammation. These results contrast sharply with those of our own study, which reported lower FENO values. However, due to the differing methodologies employed by Radhakrishnan et al, drawing a definitive comparison may not be feasible.

Our study holds certain notable advantages. This study represents a distinctive investigation within a population where SCD is relatively common, yet this specific aspect of the condition has not been thoroughly examined in the area. Additionally, in contrast to prior research, we focused on recruiting younger children with SCD, in whom PFT pattern disruptions are apparent, thereby highlighting the necessity for early pulmonary function screening in this demographic. Furthermore, our multivariate analysis revealed that an earlier onset of the pulmonary component of the disease is associated with greater severity compared to later onset cases. Finally, we observed that children with SCD exhibited lower levels of fractional exhaled nitric oxide (FeNO), which did not show a correlation with disease severity.

Our study had several limitations as well. We had excluded children using inhaled bronchodilators and those with any acute or chronic cardiorespiratory conditions. Those are important patients to potentially consider as they may represent those with more severe pulmonary disease and would possibly change the outcome of the study. It should be emphasized that PFTs are highly dependent on timing and circumstances, thus readings on a single day may not provide a comprehensive assessment. The same holds true for FeNO. Moreover it will not be unfair to assume that older children would have performed the tests with more comprehension, even though the tests met acceptability and reproducibility prior to being included in this analysis. The cognitive demands of the PFT prevented us from evaluating the pulmonary function of children younger than six years old. Moreover, we observed a small effect size, indicating the need for a more comprehensive study. Additionally, we were unable to find a relationship between the PFT outcomes and the episodes of ACS, as these episodes were not recorded. The sample size was determined based on convenience, as time constraints posed a significant limitation.

We intend to expand our research in the area of hemolysis leading to pulmonary abnormalities to gain a deeper understanding of the pathophysiology underlying the obstructive pattern in PFTs among individuals with SCD. In future investigations, we plan to utilize oscillometry to identify these dysfunctions at an earlier stage and possibly evaluate alveolar NO along with FeNO for a more comprehensive understanding.

## Conclusions

Pulmonary function deficiencies in the form of restrictive, obstructive or mixed patterns are frequently observed in children diagnosed with SCD, with restrictive and mixed patterns being particularly prevalent. Further validation through definitive assessments, such as diffusion studies and lung-volume studies, is necessary. Additionally, FeNO levels do not correlate with the severity of PFT abnormalities.

## Acknowledgments

We are grateful to the children and parents of children who participated in our research.

## Author contributions

**Conceptualization:** Gaurav Sarnaik, Atul Jindal.

**Data curation:** Gaurav Sarnaik, Nikita Agarwal, Atul Jindal.

**Formal analysis:** Gaurav Sarnaik, Nikita Agarwal, Atul Jindal.

**Investigation:** Gaurav Sarnaik.

**Methodology:** Abinaya Kannan, Gaurav Sarnaik, Nikita Agarwal.

**Project administration:** Gaurav Sarnaik, Atul Jindal.

**Software:** Abinaya Kannan, Nikita Agarwal.

**Supervision:** Atul Jindal.

**Validation:** Abinaya Kannan.

**Visualization:** Abinaya Kannan, Nikita Agarwal.

**Writing – original draft:** Abinaya Kannan, Nikita Agarwal.

**Writing – review & editing:** Atul Jindal.

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
