## [Decision Letter · Decision Letter 0]

PONE-D-25-10875The Sickened Lungs of Sickle Cell Disease: is it Restrictive or Obstructive?: A Case-Control StudyPLOS ONE

Dear Dr. Jindal,

Thank you for submitting your manuscript to PLOS ONE. After careful consideration, we feel that it has merit but does not fully meet PLOS ONE’s publication criteria as it currently stands. Therefore, we invite you to submit a revised version of the manuscript that addresses the points raised during the review process. Areas to focus the revision on include a better description of the methodology used to define restrictive and obstructive lung disease and for the FeNO measurements.  

We look forward to receiving your revised manuscript.

Kind regards,

Santosh L. Saraf

Academic Editor

PLOS ONE

Journal Requirements:

4. In the online submission form, you indicated that data is available with the corresponding author upon reasonable request.

5. Please remove all personal information, ensure that the data shared are in accordance with participant consent, and re-upload a fully anonymized data set.

Reviewers' comments:

Reviewer's Responses to Questions

**Comments to the Author**

1. Is the manuscript technically sound, and do the data support the conclusions?

Reviewer #1: Partly

Reviewer #2: Partly

2. Has the statistical analysis been performed appropriately and rigorously? 

Reviewer #1: No

Reviewer #2: I Don't Know

3. Have the authors made all data underlying the findings in their manuscript fully available?

Reviewer #1: No

Reviewer #2: No

4. Is the manuscript presented in an intelligible fashion and written in standard English?

Reviewer #1: Yes

Reviewer #2: No

5. Review Comments to the Author

Reviewer #1: This case-control study investigates pulmonary function and fractional exhaled nitric oxide (FeNO) levels in children and adolescents (aged 6–20 years) with stable sickle cell disease (SCD) compared to healthy controls. The study aims to understand the evolving pattern of lung function in children and adolescents with SCD and explore the potential role of FeNO as a biomarker for airway inflammation and a predictor of acute chest syndrome (ACS) risk. The measurement of FeNO adds a novel dimension to the study, linking inflammation (as reflected by FeNO) to pulmonary function in SCD.

Introduction:

75-78: Nitric oxide (NO) is produced by red blood cells (RBCs) and is crucial for the process of

vasodilation. In sickle cell disease (SCD), the functionality of RBCs is compromised, which

adversely affects NO activity. This impairment contributes to the exacerbation of symptoms

during vaso-occlusive crises, creating a detrimental cycle. [8]

The authors may do well to highlight the different cellular sources of NO and the primary contributor to the NO measured as FeNO.

Methods:

FeNO Measurement:

While FeNO is a valuable biomarker for airway inflammation, the methodology does not explicitly describe how FeNO levels were measured or standardized. Clarifying the protocol (e.g., device used, number of measurements, exhalation flow rate) would enhance reproducibility.

Lack of Blinding:

The study does not mention whether researchers conducting PFTs or analyzing data were blinded to the case/control status. Lack of blinding could introduce observer bias.

Adjust for Multiple Comparisons:

Use corrections for multiple comparisons (e.g., Bonferroni) to reduce the risk of Type I errors.

Results:

The study reports a weak positive correlation between hydroxyurea duration and disease severity, but this could reflect confounding by indication (i.e., more severe cases are more likely to receive hydroxyurea).

Instead of multiple correlation analysis, the authors could consider a multivariate analysis like regression to control for other variables.

Discussions:

FeNO as a Biomarker:

While FeNO is a useful biomarker for Th2-mediated eosinophilic airway inflammation, its role in SCD-related lung disease is not well-defined. The study assumes FeNO reflects general inflammation, which may not be accurate. Additional biomarkers (e.g., sputum eosinophils, CRP, IL-6) could provide a more comprehensive assessment of airway inflammation.

The authors initially suspected that FeNO levels would be higher in SCD patients compared to controls, due to their known respiratory issues, general inflammation, and that FeNO could potentially be use as a diagnostic and treatment monitoring tool like in eosinophilic asthma. However, their findings did not support this. Authors continue to highlight that this may be because of NO depletion in SCD. However, The primary source of elevated NO in asthma is iNOS, which is upregulated in airway epithelial cells and inflammatory cells (e.g., eosinophils, macrophages) in response to inflammation. Authors should discuss potential reasons for conflicting FeNO results across multiple studies, such as differences in patient populations, disease states, or measurement techniques.

Lines 254-256: Thirdly, our analysis revealed that a younger

onset of the disease is associated with increased severity and a mixed or restrictive pattern

rather than an obstructive one, a finding that has not been previously reported.

It’s unclear what younger onset of disease means as all patients had SCD at birth.

Lastly, we examined the correlation between elevated FeNO levels and the frequency of blood

transfusions, thereby elucidating the underlying pathophysiological mechanisms at the level

of red blood cells.

The authors are not clear on how blood transfusion history was obtained. It would be nice to control for blood transfusions or give FeNO values for transfused vs transfused cases.

Tables/Figures need description

Figure 2 needs a legend

Reviewer #2: This is important work in the field of sickle cell lung disease. We need more studies looking at pulmonary function testing in these patients to provide more data detailing the abnormalities we see with the hopes of eventually improving screening, identification of disease, and treatment. There are, however, some major concerns I have with some of the fundamental terminology and methodology of the paper, and conclusions that this paper draws from the results they obtained.

Major comments:

1. I am concerned about the use of the term restrictive in the title and throughout the paper. Restrictive lung disease is only able to be diagnosed by obtaining lung volumes, which was not done here. A restrictive pattern can be suggested by spirometry but to truly diagnose restriction it needs to be confirmed with lung volumes, and often a “restrictive pattern” does not end up leading to a true restriction diagnosis when you obtain lung volumes. It is often difficult to say what a restrictive pattern means clinically in general, and especially for these patients. Based on what I know of sickle cell lung disease and prior work in the field, I would also not expect true restrictive lung disease to be present in earlier age groups. This prior work includes articles you cite [3-5] with lower TLC values and increased restriction diagnosed with older age, and citing other articles that say the same. Specifically, in [5] by Arteta they note, “We categorized pulmonary function only in participants who performed both spirometry and plethysmography due to the inability of the former to identify restrictive lung function accurately. For example, six of 14 patients in our cohort with an apparently restrictive finding on spirometry (low FVC and FEV1 and normal FEV1/FVC) had a normal TLC.” I would recommend changing the title of the paper and keywords because we aren’t talking about diagnosed restrictive lung disease and that is what is implied. I would also change anywhere in the paper that implies a restrictive lung disease or mixed defect diagnosis instead of a pattern. This should also be mentioned as a limitation in the discussion. Here is a recent review article detailing more about diagnosing ventilatory defects and has a nice section about restriction if it is helpful: (Barkous B, Briki C, Boubakri S, Abdesslem M, Ben Abbes N, Ben Hmid W, Ben Saad H. Routine pulmonary lung function tests: Interpretative strategies and challenges. Chron Respir Dis. 2024 Jan-Dec;21:14799731241307252. doi: 10.1177/14799731241307252. PMID: 39644209; PMCID: PMC11625406.)

2. I would recommend adding if all tests met acceptability and reproducibility prior to being included in this study/analysis. For example, in [5] Arteta that you cite they write, “Results were printed in a standardized manner and reviewed by the same investigator (M.A.) to ensure acceptability and reproducibility according to ATS/ERS standards before including them in the analysis”, and in [3] Al Biltagi they write, “At least three acceptable and reproducible maneuvers were performed and the best values were considered.” Patients of any age are not always able to produce results that meet acceptability and reproducibility criteria. This is especially important when considering including younger patients. The cohort you used includes patients who are 6-7 years old, and often patients this young are not able to meet acceptability and reproducibility criteria for testing to include in analysis like this. If this was not done, this is a significant limitation and should be listed as a limitation in the discussion.

3. I am confused on what you are using to define obstruction. In the second paragraph of your methodology (lines 114-116), you say an obstructive defect is characterized by a reduced FEV1/FVC ratio of <70% or below the 5th percentile for adolescents (in the paper you cite [11] this says “adults” not “adolescents” so I would make sure to correct that, along with the restrictive pattern FVC <5th percentile (lines 117-118) in “adults” not “adolescents”), and <85% for ages 5-18 years. It is not clear to me which of these numbers you used (<70%, <5th percentile, or <85%) if you could clarify, especially since the age range you chose is 6-20 yo so includes patients in both age groups <18 years old and >18 years old.

Minor comments:

1. Specify in the 1st paragraph of the methodology (lines 98-99) how long the patients were in a stable state of health without pain, crises, or acute illnesses – was it one week, one month, one year?

2. As far as I’m aware, nitric oxide is not produced by red blood cells, or at most very minimally, but is instead activated. Would change the wording of the start of the 4th paragraph in the introduction (line 75). I would also separate the two thoughts you have in this paragraph because NO activity in the vasculature and FeNO in the airway are different and still have an unclear connection, and the relationship is even more complicated in SCD. If you want to leave them in the same paragraph, I would explain more using prior studies to talk about the possible connection. In airway inflammation FeNO is typically elevated, but in SCD and vasculature NO is typically depleted, so providing more commentary on this aspect if you are going to discuss your FeNO results would be helpful. Similarly, in the 4th paragraph of the discussion your conclusion (lines 230-233) you write, “the abnormal NO activity in sickle cell anemia red blood cells exacerbates vaso-occlusive crises, which is clinically reflected in the observed low FeNO levels in our cohort compared to healthy children.” Since you did not collect information on VOCs in this study to compare to, I don’t think you can make this conclusion and would instead talk more about just the FeNO levels and what that could indicate. Also in this paragraph, NO in the sentence referring to the Girgis study [18] (line 223) should be FeNO.

3. In the 7th paragraph of the results (lines 161-163) there is interpretation of the results: “this probably hints…” but interpretation of the results should be reserved for the discussion, with the results section simply detailing the results of your analysis.

4. In the 2nd paragraph of your discussion, you say 74% of patients from Hulke [13] exhibited a restrictive pattern, but in reviewing the paper that number is 24% (line 186). I am also not sure how your findings stand in contrast, especially since the studies are in different age groups, so I would highlight that and discuss more.

5. I am having trouble connecting most of the 3rd paragraph of your discussion (lines 198-214) back to the paper – including the sentence on LDH levels. I am not entirely sure why you chose the include studies that you did, and how they relate to what your study looks at. Would recommend connecting this paragraph and the studies that you mention back to relevance and findings from your study.

6. In the 7th paragraph of the discussion, I’m confused by the sentence, “Secondly, we successfully utilized fractional exhaled nitric oxide (FeNO) as a proxy for obstructive dysfunction, which corroborated our findings, as we observed a predominance of restrictive patterns over obstructive ones” (lines 252-254). I don’t know where that comes from in your results to use FeNO as a proxy for obstruction, and I’m not sure how that would relate to a predominance of restrictive patterns. Please clarify.

7. Try to organize the discussion in a more logical way. For example, put all limitations together. You mention this was a non-funded study with limited time and possible sampling bias in the first paragraph, but more of the limitations are listed later in the discussion. Also, try to incorporate your study results with the papers that you discuss as mentioned above to connect your study more with current literature.

8. Another limitation you should mention in your discussion is the exclusion of children who were being treated with inhaled bronchodilators, since this group may have represented even more severe pulmonary disease.

9. Since you don’t talk about incentive spirometry at all in the paper I wouldn’t bring it in for the last sentence of your discussion (lines 265-266) since you haven’t established how that would related to your findings. It might be better to say you would want to connect the parameters that you studied with respiratory symptoms and other clinical markers of disease, or something else.

10. I would read through again and make sure to edit for grammar since there are quite a few errors throughout. For example, in the last paragraph of the results the sentence, “more the number of blood transfusions, lesser were the FeNO levels” does not make sense as it is written. You also need to make sure that all abbreviations are written out originally, and then you don’t need to re-define them later. For example, in the introduction PFTs, ACS, and Th2 are not defined before they are used as abbreviations, but FeNO is defined twice and SCD defined three times. In the 3rd paragraph of the methodology, you repeat FEV1 in “FEV1 percent of predicted FEV1” that only needs to say FEV1 once.

11. Citations should be moved to the end of the sentence(s) that they connect with, not the beginning of the sentence.

6. PLOS authors have the option to publish the peer review history of their article (what does this mean? ). If published, this will include your full peer review and any attached files.

**Do you want your identity to be public for this peer review?** For information about this choice, including consent withdrawal, please see our Privacy Policy .

Reviewer #1: No

Reviewer #2: No

---

## [Author Response · Author response to Decision Letter 1]

23 Apr 2025

Dear reviewers

We are grateful for your time and effort in reviewing out manuscript. Hereby we attach the response to your comments.

1 Please ensure that your manuscript meets PLOS ONE's style requirements, including those for file naming. The PLOS ONE style templates can be found at https://journals.plos.org/plosone/s/file?id=wjVg/PLOSOne_formatting_sample_main_body.pdf and https://journals.plos.org/plosone/s/file?id=ba62/PLOSOne_formatting_sample_title_authors_affiliations.pdf

Response

Respnse

Made necessary changes in accordance to PLOS One guidelines

Comment

Your ethics statement should only appear in the Methods section of your manuscript. If your ethics statement is written in any section besides the Methods, please move it to the Methods section and delete it from any other section. Please ensure that your ethics statement is included in your manuscript, as the ethics statement entered into the online submission form will not be published alongside your manuscript

Response

Made the necessary changes according to your guidance

Comment

We note that you have indicated that there are restrictions to data sharing for this study. For studies involving human research participant data or other sensitive data, we encourage authors to share de-identified or anonymized data. However, when data cannot be publicly shared for ethical reasons, we allow authors to make their data sets available upon request. For information on unacceptable data access restrictions, please see http://journals.plos.org/plosone/s/data-availability#loc-unacceptable-data-access-restrictions

Response

We have uploaded the anonymised data

Comment

he authors may do well to highlight the different cellular sources of NO and the primary contributor to the NO measured as FeNO

Response

We have changed the introduction majorly to accommodate the requirements

Comment

While FeNO is a valuable biomarker for airway inflammation, the methodology does not explicitly describe how FeNO levels were measured or standardized. Clarifying the protocol (e.g., device used, number of measurements, exhalation flow rate) would enhance reproducibility

Response

Clarification to the protocol added in the methods

Comment

The study does not mention whether researchers conducting PFTs or analyzing data were blinded to the case/control status. Lack of blinding could introduce observer bias.

Response

Required changes made in the methodology

Comment

Instead of multiple correlation analysis, the authors could consider a multivariate analysis like regression to control for other variables.

Response

Analysis done and added to the results

Comment

Authors should discuss potential reasons for conflicting FeNO results across multiple studies, such as differences in patient populations, disease states, or measurement techniques

Response

We have deeply examined this issue and mentioned in the revised discussion

Comment

It’s unclear what younger onset of disease means as all patients had SCD at birth.

Response

We identify the problem in the statement and we have corrected ourselves

Comment

The authors are not clear on how blood transfusion history was obtained. It would be nice to control for blood transfusions or give FeNO values for transfused vs transfused cases.

Response

We have corrected this problem in the methods section

Comment

Restrictive lung disease is only able to be diagnosed by obtaining lung volumes, which was not done here. A restrictive pattern can be suggested by spirometry but to truly diagnose restriction it needs to be confirmed with lung volumes, and often a “restrictive pattern” does not end up leading to a true restriction diagnosis when you obtain lung volumes

Response

We identified the error made by us. We have therefore replaced restrictive disease with pattern in all of the manuscript wherever necessary

Comment

I would recommend changing the title of the paper and keywords because we aren’t talking about diagnosed restrictive lung disease and that is what is implied. I would also change anywhere in the paper that implies a restrictive lung disease or mixed defect diagnosis instead of a pattern. This should also be mentioned as a limitation in the discussion.

Response

Point well taken and title changed

Comment

I would recommend adding if all tests met acceptability and reproducibility prior to being included in this study/analysis. Response

Amendment made in the methods section

Comment

I am confused on what you are using to define obstruction. In the second paragraph of your methodology (lines 114-116), you say an obstructive defect is characterized by a reduced FEV1/FVC ratio of <70% or below the 5th percentile for adolescents (in the paper you cite [11] this says “adults” not “adolescents” so I would make sure to correct that, along with the restrictive pattern FVC <5th percentile (lines 117-118) in “adults” not “adolescents”), and <85% for ages 5-18 years. It is not clear to me which of these numbers you used (<70%, <5th percentile, or <85%) if you could clarify, especially since the age range you chose is 6-20 yo so includes patients in both age groups <18 years old and >18 years old.

Response

Made the changes in the methods and have clarified the language

Comment

Specify in the 1st paragraph of the methodology (lines 98-99) how long the patients were in a stable state of health without pain, crises, or acute illnesses – was it one week, one month, one year?

Response

Corrections made

Comment

In the 7th paragraph of the results (lines 161-163) there is interpretation of the results: “this probably hints…” but interpretation of the results should be reserved for the discussion, with the results section simply detailing the results of your analysis.

4. In the 2nd paragraph of your discussion, you say 74% of patients from Hulke [13] exhibited a restrictive pattern, but in reviewing the paper that number is 24% (line 186). I am also not sure how your findings stand in contrast, especially since the studies are in different age groups, so I would highlight that and discuss more.

5. I am having trouble connecting most of the 3rd paragraph of your discussion (lines 198-214) back to the paper – including the sentence on LDH levels. I am not entirely sure why you chose the include studies that you did, and how they relate to what your study looks at. Would recommend connecting this paragraph and the studies that you mention back to relevance and findings from your study.

6. In the 7th paragraph of the discussion, I’m confused by the sentence, “Secondly, we successfully utilized fractional exhaled nitric oxide (FeNO) as a proxy for obstructive dysfunction, which corroborated our findings, as we observed a predominance of restrictive patterns over obstructive ones” (lines 252-254). I don’t know where that comes from in your results to use FeNO as a proxy for obstruction, and I’m not sure how that would relate to a predominance of restrictive patterns. Please clarify.

7. Try to organize the discussion in a more logical way. For example, put all limitations together. You mention this was a non-funded study with limited time and possible sampling bias in the first paragraph, but more of the limitations are listed later in the discussion. Also, try to incorporate your study results with the papers that you discuss as mentioned above to connect your study more with current literature.

8. Another limitation you should mention in your discussion is the exclusion of children who were being treated with inhaled bronchodilators, since this group may have represented even more severe pulmonary disease.

9. Since you don’t talk about incentive spirometry at all in the paper I wouldn’t bring it in for the last sentence of your discussion (lines 265-266) since you haven’t established how that would related to your findings. It might be better to say you would want to connect the parameters that you studied with respiratory symptoms and other clinical markers of disease, or something else.

10. I would read through again and make sure to edit for grammar since there are quite a few errors throughout. For example, in the last paragraph of the results the sentence, “more the number of blood transfusions, lesser were the FeNO levels” does not make sense as it is written. You also need to make sure that all abbreviations are written out originally, and then you don’t need to re-define them later. For example, in the introduction PFTs, ACS, and Th2 are not defined before they are used as abbreviations, but FeNO is defined twice and SCD defined three times. In the 3rd paragraph of the methodology, you repeat FEV1 in “FEV1 percent of predicted FEV1” that only needs to say FEV1 once.

11. Citations should be moved to the end of the sentence(s) that they connect with, not the beginning of the sentence.

Response

We have revised the entire discussion as the previous one was deeply flawed.

---

## [Decision Letter · Decision Letter 1]

PONE-D-25-10875R1“The Pattern of Lung Function Tests in Children with Sickle Cell Disease: A Case-Control Study”PLOS ONE

Dear Dr. Jindal,

Thank you for submitting your manuscript to PLOS ONE. After careful consideration, we feel that it has merit but does not fully meet PLOS ONE’s publication criteria as it currently stands. Therefore, we invite you to submit a revised version of the manuscript that addresses the points raised during the review process. Please pay careful attention to address all of the grammatical errors cited by the reviewer as well as issues with the tables (e.g. one table is missing, continuous vs. categorical variables) and clarify the discussion.

We look forward to receiving your revised manuscript.

Kind regards,

Santosh L. Saraf

Academic Editor

PLOS ONE

Journal Requirements:

Reviewers' comments:

Reviewer's Responses to Questions

**Comments to the Author**

1. If the authors have adequately addressed your comments raised in a previous round of review and you feel that this manuscript is now acceptable for publication, you may indicate that here to bypass the “Comments to the Author” section, enter your conflict of interest statement in the “Confidential to Editor” section, and submit your "Accept" recommendation.

Reviewer #1: All comments have been addressed

2. Is the manuscript technically sound, and do the data support the conclusions?

Reviewer #1: Yes

3. Has the statistical analysis been performed appropriately and rigorously? 

Reviewer #1: I Don't Know

4. Have the authors made all data underlying the findings in their manuscript fully available?

Reviewer #1: No

5. Is the manuscript presented in an intelligible fashion and written in standard English?

Reviewer #1: No

6. Review Comments to the Author

Reviewer #1: Abstract

Background

• "definitive change" is vague. Suggest using "widespread involvement."

• Consider linking inflammation and lung dysfunction more explicitly.

Methods

• Define devices more clearly; “hand-held device” is vague.

• “Data was collected and analysed.” → “Data were collected and analyzed.”

Results

• CI for FEV1/FVC appears to have an error: duplicate "95% CI."

• Better organization would improve readability: group findings (e.g., PFT differences, clinical correlations).

Example:

Results

Children with SCD had significantly lower pulmonary function values compared to controls:

• FEV1 median difference: 33.5% (95% CI: 27.2–38.0; p<0.0001)

• FVC: 25.4% (95% CI: 30.0–32.25; p<0.0001)

• FEV1/FVC: 0.088 (95% CI: 0.075–0.083; p<0.0001)

• PEFR: 24.8% (95% CI: 16.38–33.8; p<0.0001)

• FeNO: 8.17 ppb (95% CI: 5.77–12.65; p<0.0001)

Pulmonary function abnormalities were associated with younger age (p=0.0022). Age (p=0.0075) and weight (p=0.0093) were significantly associated with PFT severity, while blood transfusion frequency, hydroxyurea duration, and FeNO levels were not.

Conclusions

• "FeNO levels does not..." → should be "do not."

• Clarify "severity of illness" – based on PFT pattern?

Introduction:

• Clarity & Grammar:

• "individuals SCD" should be "individuals with SCD."

• The phrase "mainly found in the lower parts of the lungs" is informal—could be more technical.

• Better explain study aim

Results:

Table 5 is missing.

Table 6 implies that the dependent variable was a continuous variable and not the categorical severity pattern shown in Table 4.. The authors do not clarify what this measure of severity was, FEV1?

interpretation of the analysis.

Take for instance, Age, on face value, one would interpret it as follows:

Coefficient: -0.0658

• Interpretation: Older age is associated with worsening of the dependent variable (lower PFT scores), and this is statistically significant (p = 0.00749).

• Clinical Insight: Lung impairment may worsen with age in SCD

However, the authors in the discussion now state: "It was noted that older age was significantly linked to either a normal or obstructive pattern. In the multivariate analysis, weight and age exhibited a negative correlation with the severity of illness."

It�s important to emphasize the results of the multivariate analysis and downplay the findings from simple comparisons, especially when they conflict with the multivariate analysis.

The phrase "risk difference in our cohort of patient is 0.83" needs clarification—what risk is being referenced?

Tables and figures are labeled, but data is hard to follow due to repeated or inconsistent labels.

Mentioning the Bonferroni correction is excellent, but more context is needed: how many comparisons were made to justify the adjusted α?

Discussion:

• Strengthen the connection between multivariate regression and the discussion

While your regression analysis showed significant effects of age and weight, the discussion does not fully integrate these findings with the broader conclusions. A more explicit link between the statistical results and disease severity would improve coherence.

•There is too much discussion on unrelated studies. …Kassim …can be removed

Lines 230: We however did not do definitive diagnosis for restriction with diffusion studies, so the findings of our study are mainly ‘pattern’-based.

Lung restriction is diagnosed primarily by lung volume measurements. DLCO helps characterize restrictive patterns but cannot diagnose restriction on its own.

•

Discussion can be more concise.

7. PLOS authors have the option to publish the peer review history of their article (what does this mean? ). If published, this will include your full peer review and any attached files.

**Do you want your identity to be public for this peer review?** For information about this choice, including consent withdrawal, please see our Privacy Policy .

Reviewer #1: No

---

## [Author Response · Author response to Decision Letter 2]

20 May 2025

S. No.

1 • "definitive change" is vague. Suggest using "widespread involvement."

Reply: Thank you for wording it correctly; the change has been made in the unmarked copy at 64

2 • Consider linking inflammation and lung dysfunction more explicitly

Reply: Thank you for your observation, however due to limitation of words, we are unable to do so in the abstract, we however go into the details in the main article

3 • Define devices more clearly; “hand-held device” is vague.

Reply: In the unmarked copy line 72, we mention the name of the device.

4 • “Data was collected and analysed.” → “Data were collected and analyzed.”

Reply: Made the necessary change at line 72. Thank you for pointing out the obvious mistake.

5 • CI for FEV1/FVC appears to have an error: duplicate "95% CI."

• Better organization would improve readability: group findings (e.g., PFT differences, clinical correlations).

Example:

Results

Children with SCD had significantly lower pulmonary function values compared to controls:

• FEV1 median difference: 33.5% (95% CI: 27.2–38.0; p<0.0001)

• FVC: 25.4% (95% CI: 30.0–32.25; p<0.0001)

• FEV1/FVC: 0.088 (95% CI: 0.075–0.083; p<0.0001)

• PEFR: 24.8% (95% CI: 16.38–33.8; p<0.0001)

• FeNO: 8.17 ppb (95% CI: 5.77–12.65; p<0.0001)

Pulmonary function abnormalities were associated with younger age (p=0.0022). Age (p=0.0075) and weight (p=0.0093) were significantly associated with PFT severity, while blood transfusion frequency, hydroxyurea duration, and FeNO levels were not.

Reply: Thank you for suggesting a more robust method of representing out results. We have done the same in the abstract from line 73 to 83

6 • "FeNO levels does not..." → should be "do not."

Reply: Made the change at line 85

7 • "individuals SCD" should be "individuals with SCD."

Reply: Made the change at line 94

8 • The phrase "mainly found in the lower parts of the lungs" is informal—could be more technical

Reply: Changed at line 92-93

9 • Better explain study aim

Reply: “This study examines the PFT and FeNO results of children with SCD (ages 6 to 20 years) in comparison to age-, sex-, and height-matched individuals without SCD.”

- Lines 114 to 115

10 Table 5 is missing.

Reply: We apologize for the oversight. We have added the table

11 Table 6 implies that the dependent variable was a continuous variable and not the categorical severity pattern shown in Table 4.. The authors do not clarify what this measure of severity was, FEV1?

Reply: The initial analysis for severity was based on the FEV1% of the expected. On reviewing literature we realized that severity is based on the z-scores of FEV1, therefore we have made the change and updated the analysis

12 interpretation of the analysis.

Take for instance, Age, on face value, one would interpret it as follows:

Coefficient: -0.0658

• Interpretation: Older age is associated with worsening of the dependent variable (lower PFT scores), and this is statistically significant (p = 0.00749).

• Clinical Insight: Lung impairment may worsen with age in SCD

However, the authors in the discussion now state: "It was noted that older age was significantly linked to either a normal or obstructive pattern. In the multivariate analysis, weight and age exhibited a negative correlation with the severity of illness.”

Reply: Thank you for simplifying the method of description for us. We have made the necessary changes

13 It�s important to emphasize the results of the multivariate analysis and downplay the findings from simple comparisons, especially when they conflict with the multivariate analysis.

Reply: We identify this issue and we have made the necessary amendment

14 The phrase "risk difference in our cohort of patient is 0.83" needs clarification—what risk is being referenced?

Reply: We identified that an odd’s ratio must be calculated for a case-control study and we have made the change accordingly

15 Tables and figures are labeled, but data is hard to follow due to repeated or inconsistent labels.

Reply: We have tried to resolve this issue to the best of our abilities

16 Mentioning the Bonferroni correction is excellent, but more context is needed: how many comparisons were made to justify the adjusted α? 6 comparisons were made.

Reply: Added in the text

17 • Strengthen the connection between multivariate regression and the discussion

While your regression analysis showed significant effects of age and weight, the discussion does not fully integrate these findings with the broader conclusions. A more explicit link between the statistical results and disease severity would improve coherence.

• Discussion can be more concise.

Reply: We have tried to do so in the new discussion

18 •There is too much discussion on unrelated studies. …Kassim …can be removed

Reply: Omitted as advised

19 Lines 230: We however did not do definitive diagnosis for restriction with diffusion studies, so the findings of our study are mainly ‘pattern’-based.

Lung restriction is diagnosed primarily by lung volume measurements. DLCOhelps characterize restrictive patterns but cannot diagnose restriction on its own.

Reply: Thank you for pointing out our oversight. We have made the necessary changes.

---

## [Decision Letter · Decision Letter 2]

PONE-D-25-10875R2“The Pattern of Lung Function Tests in Children with Sickle Cell Disease: A Case-Control Study”PLOS ONE

Dear Dr. Jindal,

Thank you for submitting your manuscript to PLOS ONE. After careful consideration, we feel that it has merit but does not fully meet PLOS ONE’s publication criteria as it currently stands. Therefore, we invite you to submit a revised version of the manuscript that addresses the points raised during the review process. Reviewer #2, an expert in sickle cell-related lung disease has requested additional modifications, such as clarifying how restrictive and obstructive lung patterns were determined and to avoid using that terminology if the distinction is not clear.   Please submit your revised manuscript by Jul 27 2025 11:59PM. If you will need more time than this to complete your revisions, please reply to this message or contact the journal office at plosone@plos.org . Please include the following items when submitting your revised manuscript:

We look forward to receiving your revised manuscript.

Kind regards,

Santosh L. Saraf

Academic Editor

PLOS ONE

Reviewers' comments:

Reviewer's Responses to Questions

**Comments to the Author**

1. If the authors have adequately addressed your comments raised in a previous round of review and you feel that this manuscript is now acceptable for publication, you may indicate that here to bypass the “Comments to the Author” section, enter your conflict of interest statement in the “Confidential to Editor” section, and submit your "Accept" recommendation.

Reviewer #1: All comments have been addressed

Reviewer #2: (No Response)

2. Is the manuscript technically sound, and do the data support the conclusions?

Reviewer #1: Yes

Reviewer #2: Yes

3. Has the statistical analysis been performed appropriately and rigorously? 

Reviewer #1: I Don't Know

Reviewer #2: I Don't Know

4. Have the authors made all data underlying the findings in their manuscript fully available?

Reviewer #1: No

Reviewer #2: No

5. Is the manuscript presented in an intelligible fashion and written in standard English?

Reviewer #1: Yes

Reviewer #2: No

6. Review Comments to the Author

Reviewer #1: (No Response)

Reviewer #2: There have been significant improvements made to the paper. I still think this is a major revision since the number of errors throughout the paper in grammar, abbreviations, lower case vs upper case use, and citation placement distracted from my ability to read the paper easily for content, and I often had to re-read the same sentence or paragraph a few times to try to understand what point was being made. I am also concerned that some of the citations that I spot checked do not seem to reference the information used. I recommend these grammatical errors and citation concerns be fixed prior to possible publication.

Here are examples of the grammatical changes that need to be made. I recommend that the paper be fully re-reviewed for all similar grammatical changes to be corrected.

1. Abbreviations should be written out when they are first encountered in the abstract and again in the paper. After that, those same abbreviations do not need to be re-defined later. For example, line 65 “PEFR” and line 84 “Th2” are not previously defined anywhere and should be written out as “peak expiratory flow rate”, and “T helper 2”, respectively. As another example, lines 81-82 “pulmonary function tests” should have “(PFTs)” after it but then not later in the paper, such as in line 171, 197, and 227, and the abbreviation “PFT” should be used like in line 206. This also occurs with defining “sickle cell disease (SCD)”, “lactate dehydrogenase (LDH)”, “hemoglobin (Hb)”, “nitric oxide (NO)”, and others.

2. Use appropriate upper and lower case. For example, line 75 “Sickle Cell Disease”, and lines 77-78 “Vaso-Occlusive Crises” and “Acute Chest Syndrome”, should all be lower case.

3. Citations should be moved inside of the period. For example, line 77 should be “worldwide [1].”

4. Throughout the paper there are a lot of places where there should be a space after a period before the start of the next sentence. Specifically, line 190 there needs to be a space between “in” and “dependent” because otherwise it reads “independent”, which is a different type of variable. It also should be “in the dependent” variable.

5. Lines 137-138 are awkwardly phrased. You could change it to, “Information on the history of blood transfusions and the duration of hydroxyurea administration was gathered from medication records for the cases.”

6. Line 151 write out “eight” instead of “8”.

7. Lines 191 and 204 FEV1 z-scores needs a dash, and “better” should be changed to something like “higher” since you are not discussing better or worse here but instead if they are higher or lower. Line 204 also should be “for longer” instead of “for more”.

8. Line 223 should be “Research conducted…” or “A research study conducted…” instead of “A research conducted”.

9. Line 274 “same holds true for FeNO” is not complete - recommend changing the sentence before to “PFTs and FeNO” or add “The” before “same”.

10. Line 274 “it would not be unfair to assume” instead of “it will not be unjust to assume”.

Other comments:

1. In the abstract lines 56-57 “acute or chronic cardiorespiratory conditions” should also be listed as an exclusion since you list that in lines 114-115.

2. In line 77 specify “acute conditions that occur in patients with SCD”

3. I am not able to find the information you discuss in lines 78-82 in the papers that you cite [4,5]. The definition of sickle cell chronic lung disease has evolved significantly since the Powars paper was published in 1988 and there is no one true entity “sickle cell chronic lung disease”. You could look at a more recent review such as “Updates in Pediatric Sickle Cell Lung Disease” by Gillespie published 2024 in Clin Chest Med. Additionally, in line 81, it is unclear if you mean 90% of children with SCD or of children with sickle cell chronic lung disease.

4. I am unsure why you are mentioning the sentence starting with “Consequently, markers…” lines 91-93 since you do not go on to discuss arginine in the paper again. Would recommend removing this.

5. In lines 118-119 I would recommend not mentioning specific limitations until the discussion and would move this down to the paragraph starting line 272. Instead, for lines 118-119, you could write something like, “The sample size was determined based on recruitment ability within the timeframe of the study”.

6. Line 130 would not use “diagnosed with” but instead “characterized as having”

7. Line 150 recommend “were excluded”, instead of “had to be excluded”.

8. Line 159 recommend “group with patients diagnosed with SCD”, instead of “sickle cell group”.

9. Line 182 “The heat chart…” should be deleted since that sentence is copied twice in a row.

10. Line 212 specify “abnormal findings in this study” if that is what you are referring to.

11. Lines 221 and 286 these are “lung volume” studies that would diagnose restrictive lung disease, not “diffusion” studies

12. Line 224 need to clarify “predominant abnormality” in who? Patients with SCD?

13. Line 227 instead of “them” should say “this study by Arteta”.

14. Lines 228-229 I’m not sure that we can declare that this finding supports Arteta’s hypothesis that hemolysis may be linked to the obstructive pattern just because it is occurring in older patients.

15. Line 237 clarify if you are talking about your own study again with the sentence “additionally…”. I am also unsure how this does not align with Adekile’s hypothesis in the following sentence. I re-read this paragraph a few times and I am still unsure of what is intended. Recommend re-writing these few sentences to be clearer.

16. Line 269 confusing what you mean by “earlier onset of the disease” since SCD is a genetic condition, so all patients have it at birth. Please clarify.

17. Another limitation that should be included in the discussion paragraph starting line 272 is excluding children using inhaled bronchodilators and those with any acute or chronic cardiorespiratory conditions identified. Those are important patients to potentially consider as they may represent those with more severe pulmonary disease and would possibly change the outcome of the study.

18. Would recommend putting all future directions in the same paragraph below in 280-282 taken from lines 229-231 and 240-241.

7. PLOS authors have the option to publish the peer review history of their article (what does this mean? ). If published, this will include your full peer review and any attached files.

**Do you want your identity to be public for this peer review?** For information about this choice, including consent withdrawal, please see our Privacy Policy .

Reviewer #1: No

Reviewer #2: No

---

## [Author Response · Author response to Decision Letter 3]

20 Jun 2025

1. Abbreviations should be written out when they are first encountered in the abstract and again in the paper. After that, those same abbreviations do not need to be re-defined later. For example, line 65 “PEFR” and line 84 “Th2” are not previously defined anywhere and should be written out as “peak expiratory flow rate”, and “T helper 2”, respectively. As another example, lines 81-82 “pulmonary function tests” should have “(PFTs)” after it but then not later in the paper, such as in line 171, 197, and 227, and the abbreviation “PFT” should be used like in line 206. This also occurs with defining “sickle cell disease (SCD)”, “lactate dehydrogenase (LDH)”, “hemoglobin (Hb)”, “nitric oxide (NO)”, and others.

Response: We have reviewed the entire document and we have tried to stick to this format. We are indebted to you for pointing out these minute errors

2. Use appropriate upper and lower case. For example, line 75 “Sickle Cell Disease”, and lines 77-78 “Vaso-Occlusive Crises” and “Acute Chest Syndrome”, should all be lower case.

Response: We have made these changes

3. . Citations should be moved inside of the period. For example, line 77 should be “worldwide [1].”

Response: We have made the changes throughout the document

4. Throughout the paper there are a lot of places where there should be a space after a period before the start of the next sentence. Specifically, line 190 there needs to be a space between “in” and “dependent” because otherwise it reads “independent”, which is a different type of variable. It also should be “in the dependent” variable.

Response: Made the necessary changes

5. Lines 137-138 are awkwardly phrased. You could change it to, “Information on the history of blood transfusions and the duration of hydroxyurea administration was gathered from medication records for the cases.”

Response: Made the changes

6. Line 151 write out “eight” instead of “8”.

Response: Made the changes

7. Lines 191 and 204 FEV1 z-scores needs a dash, and “better” should be changed to something like “higher” since you are not discussing better or worse here but instead if they are higher or lower. Line 204 also should be “for longer” instead of “for more”.

Response: Made the changes

8. 8. Line 223 should be “Research conducted…” or “A research study conducted…” instead of “A research conducted”.

Response:Made the changes

9. Line 274 “same holds true for FeNO” is not complete - recommend changing the sentence before to “PFTs and FeNO” or add “The” before “same”.

Response: Made the changes

10. Line 274 “it would not be unfair to assume” instead of “it will not be unjust to assume”.

Response: Made the changes

11. In the abstract lines 56-57 “acute or chronic cardiorespiratory conditions” should also be listed as an exclusion since you list that in lines 114-115.

Response: Made the change at line 57

2. In line 77 specify “acute conditions that occur in patients with SCD”

Response: Added the conditions as ACS, VOC, aplastic and hemolytic crises

3. I am not able to find the information you discuss in lines 78-82 in the papers that you cite [4,5]. The definition of sickle cell chronic lung disease has evolved significantly since the Powars paper was published in 1988 and there is no one true entity “sickle cell chronic lung disease”. You could look at a more recent review such as “Updates in Pediatric Sickle Cell Lung Disease” by Gillespie published 2024 in Clin Chest Med. Additionally, in line 81, it is unclear if you mean 90% of children with SCD or of children with sickle cell chronic lung disease.

Response: [5, now 4] Consistent with the results of pulmonary function testing, HRCT scans revealed mild basilar focal fibrosis, with fibrosis scores trending to increase with increased pulmonary artery systolic pressures (Figures 3A and 3B).

We have updated the Powars to Gillespie and made the corresponding changes in the introduction at lines 86-89 (marked copy)

4. I am unsure why you are mentioning the sentence starting with “Consequently, markers…” lines 91-93 since you do not go on to discuss arginine in the paper again. Would recommend removing this.

Response: Duly noted and made the necessary change

5. In lines 118-119 I would recommend not mentioning specific limitations until the discussion and would move this down to the paragraph starting line 272. Instead, for lines 118-119, you could write something like, “The sample size was determined based on recruitment ability within the timeframe of the study”.

Response: Duly noted and changes made

6. Line 130 would not use “diagnosed with” but instead “characterized as having”

Response: Duly noted and changes made

7. Line 150 recommend “were excluded”, instead of “had to be excluded”.

Response: Duly noted and changes made

8. Line 159 recommend “group with patients diagnosed with SCD”, instead of “sickle cell group”.

Response: Duly noted and changes made

9. Line 182 “The heat chart…” should be deleted since that sentence is copied twice in a row.

Response: Duly noted and changes made

10. Line 212 specify “abnormal findings in this study” if that is what you are referring to.

Response: Mentioned the obstructive, restrictive or mixed patterns on PFT taken as abnormal

11. Lines 221 and 286 these are “lung volume” studies that would diagnose restrictive lung disease, not “diffusion” studies

Response: We have changed the item to lung-volume study for definitive diagnosis

12. Line 224 need to clarify “predominant abnormality” in who? Patients with SCD?

Response: We have added the clarification

13. Line 227 instead of “them” should say “this study by Arteta”.

Response: Duly noted and changes made

14 Lines 228-229 I’m not sure that we can declare that this finding supports Arteta’s hypothesis that hemolysis may be linked to the obstructive pattern just because it is occurring in older patients

Response: Duly noted and changes made

15. Line 237 clarify if you are talking about your own study again with the sentence “additionally…”. I am also unsure how this does not align with Adekile’s hypothesis in the following sentence. I re-read this paragraph a few times and I am still unsure of what is intended. Recommend re-writing these few sentences to be clearer.

Response: A similar study was done by Adekile et al on Kuwaiti children [20]. The research indicated that the correlation coefficients between FEV1 and various anthropometric measurements demonstrated a significant relationship with height (r = 0.9, p < 0.001) and weight (r = 0.8, p < 0.001). Our findings were consistent, showing that weight was notably linked to improved FEV1 z-scores. In our study children who were treated with hydroxyurea for an extended duration exhibited enhanced FEV1 z-scores. Hydroxyurea is expected to increase fetal Hb in SCD, likely reducing hemolysis [21]. This does not align with the findings by of the study Adekile et al [20] as they did not demonstrate any significant correlation with Hb, fetal Hb, reticulocytes, total bilirubin, or LDH. We, as stated earlier, strive to study the hemolysis in more details in forthcoming studies.

16. Line 269 confusing what you mean by “earlier onset of the disease” since SCD is a genetic condition, so all patients have it at birth. Please clarify.

Response: earlier onset of the pulmonary component of the disease

Another limitation that should be included in the discussion paragraph starting line 272 is excluding children using inhaled bronchodilators and those with any acute or chronic cardiorespiratory conditions identified. Those are important patients to potentially consider as they may represent those with more severe pulmonary disease and would possibly change the outcome of the study.

Response: Our study had several limitations as well. We had excluded children using inhaled bronchodilators and those with any acute or chronic cardiorespiratory conditions. Those are important patients to potentially consider as they may represent those with more severe pulmonary disease and would possibly change the outcome of the study.

Would recommend putting all future directions in the same paragraph below in 280-282 taken from lines 229-231 and 240-241.

Response: Added all future direction at the end as recommended

1. We have submitted our raw data as supplementary document for your kind perusal

2. an expert in sickle cell-related lung disease has requested additional modifications, such as clarifying how restrictive and obstructive lung patterns were determined and to avoid using that terminology if the distinction is not clear. : We mention it in the methodology section as to how we came to the conclusion of the obstructive and restrictive patterns.

---

## [Decision Letter · Decision Letter 3]

“The Pattern of Lung Function Tests in Children with Sickle Cell Disease: A Case-Control Study”

PONE-D-25-10875R3

Dear Dr. Jindal,

We’re pleased to inform you that your manuscript has been judged scientifically suitable for publication and will be formally accepted for publication once it meets all outstanding technical requirements.

Kind regards,

Santosh L. Saraf

Academic Editor

PLOS ONE

Additional Editor Comments (optional):

Reviewers' comments:

Reviewer's Responses to Questions

**Comments to the Author**

1. If the authors have adequately addressed your comments raised in a previous round of review and you feel that this manuscript is now acceptable for publication, you may indicate that here to bypass the “Comments to the Author” section, enter your conflict of interest statement in the “Confidential to Editor” section, and submit your "Accept" recommendation.

Reviewer #2: All comments have been addressed

2. Is the manuscript technically sound, and do the data support the conclusions?

Reviewer #2: Yes

3. Has the statistical analysis been performed appropriately and rigorously? 

Reviewer #2: I Don't Know

4. Have the authors made all data underlying the findings in their manuscript fully available?

Reviewer #2: Yes

5. Is the manuscript presented in an intelligible fashion and written in standard English?

Reviewer #2: Yes

6. Review Comments to the Author

Reviewer #2: I appreciate the authors making changes that I recommended. This article has significantly improved from the first submission and will be a great addition to the SCD pulmonary literature.

7. PLOS authors have the option to publish the peer review history of their article (what does this mean? ). If published, this will include your full peer review and any attached files.

**Do you want your identity to be public for this peer review?** For information about this choice, including consent withdrawal, please see our Privacy Policy .

Reviewer #2: No

---

## [Editor Report · Acceptance letter]

PONE-D-25-10875R3

PLOS ONE

Dear Dr. Jindal,

I'm pleased to inform you that your manuscript has been deemed suitable for publication in PLOS ONE. Congratulations! Your manuscript is now being handed over to our production team.

Kind regards,

on behalf of

Dr. Santosh L. Saraf

Academic Editor

PLOS ONE